# Cytoglobin Silencing Promotes Melanoma Malignancy but Sensitizes for Ferroptosis and Pyroptosis Therapy Response

**DOI:** 10.3390/antiox11081548

**Published:** 2022-08-10

**Authors:** Joey De Backer, Darko Maric, Karim Zuhra, Annemie Bogaerts, Csaba Szabo, Wim Vanden Berghe, David Hoogewijs

**Affiliations:** 1Protein Chemistry, Proteomics and Epigenetic Signaling (PPES) Research Group, Department of Biomedical Sciences, University of Antwerp, 2610 Wilrijk, Belgium; 2Section of Medicine, Department of Endocrinology, Metabolism and Cardiovascular System, University of Fribourg, 1700 Fribourg, Switzerland; 3Department of Pharmacology, Faculty of Science and Medicine, University of Fribourg, 1700 Fribourg, Switzerland; 4Plasma Lab for Applications in Sustainability and Medicine-Antwerp (PLASMANT) Research Group, Department of Chemistry, University of Antwerp, 2610 Wilrijk, Belgium

**Keywords:** cytoglobin, melanoma, ferroptosis, lipid peroxidation, cell death

## Abstract

Despite recent advances in melanoma treatment, there are still patients that either do not respond or develop resistance. This unresponsiveness and/or acquired resistance to therapy could be explained by the fact that some melanoma cells reside in a dedifferentiated state. Interestingly, this dedifferentiated state is associated with greater sensitivity to ferroptosis, a lipid peroxidation-reliant, iron-dependent form of cell death. Cytoglobin (CYGB) is an iron hexacoordinated globin that is highly enriched in melanocytes and frequently downregulated during melanomagenesis. In this study, we investigated the potential effect of CYGB on the cellular sensitivity towards (1S, 3R)-RAS-selective lethal small molecule (RSL3)-mediated ferroptosis in the G361 melanoma cells with abundant endogenous expression. Our findings show that an increased basal ROS level and higher degree of lipid peroxidation upon RSL3 treatment contribute to the increased sensitivity of CYGB knockdown G361 cells to ferroptosis. Furthermore, transcriptome analysis demonstrates the enrichment of multiple cancer malignancy pathways upon CYGB knockdown, supporting a tumor-suppressive role for CYGB. Remarkably, CYGB knockdown also triggers activation of the NOD-, LRR- and pyrin domain-containing protein 3 (NLRP3) inflammasome and subsequent induction of pyroptosis target genes. Altogether, we show that silencing of CYGB expression modulates cancer therapy sensitivity via regulation of ferroptosis and pyroptosis cell death signaling pathways.

## 1. Introduction

Cutaneous melanoma is the most aggressive form of skin cancer and incidence has risen consistently over the last few decades [1], mainly through increased exposure to ultraviolet radiation [2]. When melanoma is diagnosed in the early stages, surgical resection of the lesion leads to a 5-year survival of 99%. However, metastatic melanoma has a much less favorable prognosis, with a recorded 5-year survival of 20% [1].

In 2004, no systemic therapy available for melanoma had shown any survival benefit [3]. Since then, several immune-based therapies that mediate checkpoint inhibition have been approved for the treatment of advanced melanoma, including antibodies designed to block cytotoxic T-lymphocyte-associated protein 4 (CTLA4; ipilimumab) and programmed cell death protein 1 (PD-1; nivolumab and pembrolizumab) [4]. Melanoma has one of the highest mutational burdens among solid tumors [3]. Approximately half of all melanomas have v-Raf murine sarcoma viral oncogene homolog B (*BRAF)* mutations, and 80–90% of these mutations are a missense V600E mutation, where the wild-type amino acid 600 (a valine) is replaced by a glutamic acid residue, resulting in the constitutive activation of mitogen-activated protein kinase kinase (MEK) and extracellular signal regulated kinase (ERK) signaling [5]. Dual inhibition of this pathway using a combination of BRAF and MEK inhibitors (dabrafenib and trametinib; vemurafenib and cobimetinib; encorafenib and binimetinib) has demonstrated improved overall survival [6]. However, despite all these advances in immunotherapy and targeted therapy, there are still patients that either do not respond or develop resistance.

Recently, Tsoi et al. suggested that this unresponsiveness and/or acquired resistance to therapy could be explained by the fact that melanoma cells can exist in at least four different differentiation states [7]. Some melanoma cells predominantly reside in a dedifferentiated state, while others acquire this state in response to treatment. Interestingly, this dedifferentiated state is associated with greater sensitivity to ferroptosis. Originally, ferroptosis was identified as a unique form of cell death in mammalian cancer cells by studying the effects of the small molecule eradicator of RAS and ST (erastin), (1S, 3R)-RAS-selective lethal small molecule RSL3 and related compounds [8]. Ferroptotic cell death distinguishes itself biochemically from other types of cell death through the requirement of phospholipid peroxidation, a process reliant on iron accumulation, reactive oxygen species (ROS) and polyunsaturated fatty acid-containing phospholipids (PUFA-PLs) [9].

Cytoglobin (CYGB) is a ubiquitously expressed, hexacoordinated globin believed to be involved in the regulation of redox homeostasis. Many antioxidative functions have been assigned to CYGB, including nitric oxide dioxygenase [10], nitrite reductase [11], superoxide dismutase [12] and peroxidase activity [13]. Furthermore, CYGB expression has been found to be regulated by oxidative stress and hypoxia [14,15]. Although CYGB is expressed in many different cell types and tissues, CYGB is particularly highly enriched in pigment-producing melanocytes and frequently downregulated during melanocyte-to-melanoma transition [16]. Reduced CYGB expression through hypermethylation has also been reported in other cancer cell types, indicating that CYGB potentially serves a tumor-suppressive function [17,18,19,20,21,22].

In this study, we investigated the potential effect of CYGB on the cellular sensitivity towards RSL3-mediated ferroptosis in G361 melanoma cells that express highly abundant endogenous levels of CYGB [16]. We applied lentiviral shRNA approaches to generate G361 cells with stable knockdown of CYGB expression to measure the effects of CYGB silencing on cellular bioenergetics, redox signaling and viability via redox-sensitive fluorescent DCF and ARE-dependent luciferase assays. We further employed RNA sequencing to study the CYGB-dependent gene clusters and enriched pathways under RSL3-induced ferroptosis conditions. Collectively, our results suggest a cytoprotective role of CYGB in the cellular response to ferroptosis in melanoma via the interplay of ferroptosis and pyroptosis cell death pathways.

## 2. Materials and Methods

### 2.1. Cell Culture

G361 (ATCC CRL-1424; Manassas, VA, USA) melanoma cells were maintained in Dulbecco’s Minimum Essential Media (DMEM) (Gibco, Life Technologies; Waltham, MA, USA), containing L-glutamine, supplemented with 10% heat-inactivated fetal bovine serum (FBS, Gibco, Fisher Scientific, Waltham, MA, USA) and 1% Penicillin/Streptomycin (10,000 Units/mL P; 10,000 μg/mL S; Gibco, Life Technologies; Waltham, MA, USA). Cells were incubated in a humidified 5% CO_2_ atmosphere at 37 °C and were routinely subcultured after trypsinization.

### 2.2. Generation of Stable Knockdown and Overexpressed Cell Lines

Expression vectors encoding short hairpin RNA (shRNA) sequences targeting human CYGB in a pLKO.1-puro plasmid were purchased from Sigma-Aldrich (Burlington, MA, USA) (shCYGB: order number TRCN0000059378). Control cells (shCTR) were transfected with a non-targeting control shRNA under the control of a U6 promoter in a pLKO.1 puromycin resistance vector (Sigma-Aldrich), as described previously [23]. Viral particles were produced in HEK293T cells by co-transfection of the respective transfer vector (3 μg) with the packaging plasmids pLP1 (4.2 μg), pLP2 (2 μg) and pVSV-G (2.8 μg, all from Invitrogen) using CaCl_2_ transfection, as described before [24]. G361 cells were transduced with lentiviral-pseudotyped particles and cell pools were cultured in DMEM supplemented with 10% FBS and 1% Penicillin/Streptomycin with the appropriate antibiotic for selection.

### 2.3. Determination of Cellular Bioenergetics

Cellular bioenergetics was measured by the Extracellular Flux Analysis method. Briefly, G361, G361-shCYGB and G361-shCTR cells were seeded on XFe24-well microplates (Agilent technologies, Santa Clara, CA, USA) at 2 × 10^4^ cells per well and incubated at 37 °C and 5% CO_2_ for 24 h. The day after, cells were washed twice with DMEM at pH 7.4 supplemented with L-glutamine (2 mM, Gibco), sodium pyruvate (1 mM, Sigma-Aldrich, Burlington, MA, USA) and glucose (10 mM, Sigma-Aldrich). The plate was then incubated in a CO_2_-free incubator at 37 °C for 1 h to allow temperature and pH equilibration. The assay protocol consisted of 3 min mix, 3 min wait and 3 min measurement cycles, with measurement of basal values of oxygen consumption rate (OCR) (2 cycles), followed by injection of 1 µM oligomycin, used to evaluate the ATP generation rate (2 cycles). Afterward, 0.2 µM carbonyl cyanide-4-trifluoromethoxy phenylhydrazone (FCCP) was employed to evaluate the maximal mitochondrial respiratory capacity (2 cycles). Finally, 0.5 μM of rotenone and antimycin A was injected to inhibit the electron transport through complex I and III, respectively, aiming to detect the extra-mitochondrial OCR (2 cycles). At the end of the assay, total protein per well was measured using the Bradford reagent (BioRad, Hercules, CA, USA) and OCR values were normalized to the protein amount. Data were analyzed with Wave (v. 2.6; Agilent Technologies, Santa Clara, CA, USA) and graphed with GraphPad Prism 8 (GraphPad Software Inc.; San Diego, CA, USA).

### 2.4. H_2_DCF-DA Assay

G361-shCTR and G361-shCYGB were seeded in black, clear-bottom 24-well plates at 6 × 10^5^ cells per well and incubated with 10 μM H_2_DCF-DA for 30 min in the dark. Fluorescence was measured using a 96-well fluorometer (Infinite 200Pro, Tecan, Männedorf, Switzerland).

### 2.5. Cell Viability

Cell viability was determined using the vital dye propidium iodide (PI, Invitrogen, Waltham, MA, USA) to allow the detection of dead cells. The day before treatment, 5 × 10^4^ cells were seeded in a 24-well plate, containing complete DMEM. The next day, cells were treated with 7.5 μM RSL3 (for dose kinetics, see Appendix A) and incubated for 24 h in a humidified 5% CO_2_ atmosphere at 37 °C. Cells were collected in round-bottom polystyrene tubes (Falcon, Corning, NY, USA), washed (FACS buffer; 1× phosphate-buffered saline (PBS), 3% FBS, 1mM ethylenediaminetetraacetic acid (EDTA)) and centrifuged for 5 min at 1500 rpm before being resuspended in ice-cold PBS. PI (500 ng) was added immediately before measurement on the CytoFLEX flow cytometer (Beckman Coulter, Brea, CA, USA). Data were analyzed using FlowJo software (FlowJo, BD, Franklin Lakes, NJ, USA).

### 2.6. Lipid Peroxidation Assay

The Image-iT Lipid Peroxidation Kit (Thermo Scientific, Waltham, MA, USA) was used for the detection of lipid peroxidation in live cells through oxidation of BODIPY™ 581/591 C11 reagent, according to the manufacturer’s protocol. Briefly, 5 × 10^4^ cells were seeded in a 24-well plate, containing complete DMEM. The next day, cells were treated with 7.5 μM RSL3 and incubated for 4 h in a humidified 5% CO_2_ atmosphere at 37 °C. Thirty minutes before collection, 10 μM reagent was added. Upon oxidation by lipid hydroperoxides, the reagent displays a shift in peak fluorescence emission from ~590 nm to ~510 nm. Fluorescence from live cells shifts from red to green, providing a ratiometric (red over green signal) indication of lipid peroxidation. The more lipid peroxidation, the lower the red over green ratio will be. Fluorescence was measured using the CytoFLEX flow cytometer (Beckman Coulter, Brea, CA, USA). Cumene hydroperoxide was used as a positive control.

### 2.7. Luciferase Reporter Assays

First, 1 × 10^5^ G361 cells were transiently transfected with 250 ng ARE-driven reporter plasmid and 50 ng NRF2-expressing plasmid, in a twelve-well format, using JetOptimus (Polyplus, Illkirch-graffenstaden, Alsace, France). To control for differences in transfection efficiency and extract preparation, 25 ng pRL-SV40 *Renilla* luciferase reporter vector (Promega, Madison, WI, USA) was co-transfected. Luciferase activities of triplicate wells were determined using the Dual Luciferase Reporter Assay System (Promega, Madison, WI, USA), as described before [25]. Reporter activities were expressed as relative firefly/*Renilla* luciferase activities (R.L.U.). All reporter gene assays were performed at least 3 times independently.

### 2.8. RNA Extraction, Purification and cDNA Conversion

RNA extraction and purification was performed using a PureLink RNA Mini Kit (Invitrogen, Waltham, MA, USA), according to the manufacturer’s instructions. RNA concentration and purity was measured with an Epoch spectrophotometer (BioTek, Winooski, VM, USA) by measuring absorbance at a 260/280 nm ratio. cDNA (1 μg) was reverse-transcribed using Superscript II reverse transcriptase (Invitrogen, Waltham, MA, USA), according to the manufacturer’s protocol.

### 2.9. Real-Time Quantitative PCR

Amplification of cDNA and subsequent quantification was performed using the StepOne Real-Time PCR system (Applied Biosystems, Waltham, MA, USA) using a Power SYBR Green Master Mix (Applied Biosystems, Waltham, MA, USA). The following conditions were used during PCR: 95 °C for 10 min and 40 cycles of 95 °C for 15 s; 60 °C for 1 min. All PCR reactions were performed in duplicate for three biological replicates. Results were subsequently analyzed using qbase+ software (v3.2, Biogazelle, Zwijnaarde, Ghent, Belgium), as described before [15]. A list of used reference and target genes, together with their primer sequences, amplification efficiency and amplicon size, is given in Appendix A. All primers were manufactured and provided by Eurogentec (Seraing, Liège, Belgium).

### 2.10. Protein Extraction and Quantification

Lysis buffer, containing 10 mM Tris HCl (pH 8), 1 mM EDTA, 400 mM NaCl, 1% NP-40 and protease inhibitors (Sigma-Aldrich, Burlington, MA, USA), was used to lyse cells, as described before [23]. Lysed cells were placed on a rotating arm at 4 °C for 30 min to allow optimal performance of the lysis buffer. The suspension was subsequently sonicated for 1 min at 60 Hz to degrade any potential formed DNA aggregates. Finally, samples were centrifuged at 10,000× *g* for 15 min and the protein-containing supernatant was collected. Protein concentrations were determined using the BCA Protein Assay Kit (ThermoFisher Scientific, Waltham, MA, USA).

### 2.11. Immunoblotting

Extracted proteins for immune-based Western blotting were first separated, according to molecular weight, using sodium dodecyl sulphate polyacrylamide gel electrophoresis (SDS-PAGE) gels, followed by electrotransfer to nitrocellulose membranes (Amersham Hybond-ECL, GE Healthcare, Chicago, IL, USA), as described before [26]. Equal amounts of protein and volume were loaded onto a 12.5% polyacrylamide gel for CYGB, heme oxygenase 1 (HO-1) and NF E2-related factor 2 (NRF2). Membranes were blocked in TBS-T (Tris-buffered saline; 0.1% Tween-20), containing 5% non-fat dry milk, for 1 h at room temperature. After blocking, membranes were incubated overnight at 4 °C with primary antibodies (anti-CYGB, Proteintech, Rosemont, IL, USA; 13317-1-AP; anti-β-2-Microglobulin (B2M), Proteintech, 13511-1-AP; anti-β-Actin (ACTB), Santa Cruz, sc-47778; anti-HO-1, Proteintech, 10701-1-AP; anti-NRF2, Proteintech, 16396-1-AP). The following day, membranes were washed with TBST-T and incubated for 1 h with horseradish-conjugated secondary antibodies (anti-rabbit IgG HRP, Sigma, GENA934-1ML; anti-mouse IgG HRP, Invitrogen, 31430). The signal was revealed using ECL Prime (Amersham, GERPN2232) on an Amersham Imager 680 (GE Life Sciences; Piscataway, NJ, USA) and exported and quantified using the Image Studio™ program (LI-COR Biosciences, Lincoln, NE, USA).

### 2.12. RNA Sequencing

Total RNA sample quality was assessed with TapeStation (Agilent Technologies, Santa Clara, CA, USA) and Qubit assay (Invitrogen, Waltham, MA, USA). Total RNA samples with an RNA integrity number (RIN) > 7.0 and purity (OD_260_/OD_280_) ratio 1.8–2.2 were used for subsequent experiments. Sequence libraries were generated using the poly(A) RNA selection method and sequenced by GENEWIZ (Azenta Life Sciences, Chelmsford, MA, USA). An independent library was constructed for each of the triplicate samples. High-throughput RNA sequencing was performed with pair end 150 bp reading length on an Illumina NovaSeq 6000 (Illumina, San Diego, CA, USA) sequencer. The DESeq2 analysis was used to estimate variance–mean dependence and test for differential expression [27]. Genes with a *p*-adjusted value ≤ 0.05 were considered differentially expressed. Genes with a *p*-adjusted value ≤ 0.05 and an absolute log_2_ fold change ≥ 1 were recognized as significantly differentially expressed genes (DEGs). Fast gene set enrichment analysis (fGSEA) was performed on the complete (normalized) count data [28] using the hallmark gene sets [29]. A gene ontology (GO) enrichment analysis was performed on the DEGs by implementing the software GeneSCF (v1.1-p2). The Gene Ontology Analysis (GOA) human GO list was used to cluster the set of genes based on their biological processes and determine their statistical significance [30]. A list of genes clustered based on their gene ontologies was generated. Volcano plots were generated using the EnhancedVolcano R package to visualize the results of the differential expression analyses.

## 3. Results

### 3.1. CYGB Knockdown Influences Cellular Bioenergetics

To investigate the role of CYGB in melanoma cells, we first established an shRNA-mediated CYGB knockdown cell line (G361-shCYGB) as well as a knockdown control line (G361-shCTR). RT-PCR and immunoblotting experiments illustrated efficient knockdown of CYGB at the mRNA and protein level, respectively (Figure 1A,B).

Next, the mitochondrial oxidative phosphorylation was analyzed in G361, G361-shCTR and G361-shCYGB cells through quantification of the oxygen consumption rate (OCR) using the Seahorse Extracellular Flux Analyzer (Figure 1C–G). Knockdown of CYGB resulted in an overall decrease in OCR (Figure 1C). Besides the OCR, additional respiratory parameters were measured by using different pharmacological compounds of the mitochondrial function. Initially, the baseline cellular OCR was measured. After subtraction of the non-mitochondrial respiration, a significant reduction in basal respiration was observed in G361-shCYGB cells compared to both G361 and G361-shCTR cells (Figure 1D). To determine ATP-linked respiration, the complex V inhibitor oligomycin was used. ATP production was significantly decreased in the CYGB knockdown cell line (Figure 1E). Subsequently, the maximal respiration was measured after addition of the protonophore FCCP. Similarly, a significant reduction in the maximal respiratory rate in G361-shCYGB was observed compared to both G361 and G361-shCTR (Figure 1F). Lastly, by inhibiting complex III and I, the non-mitochondrial respiration was measured. Rotenone and antimycin A inhibition showed a significant decrease in the spare respiratory capacity (Figure 1G). Collectively, these data suggest that CYGB deficiency in G361 cells negatively affects mitochondrial oxidative phosphorylation.

### 3.2. CYGB Expression Affects ROS Homeostasis and Ferroptosis Sensitivity

A potential effect of CYGB knockdown in G361 melanoma cells on the intracellular ROS levels was assessed using the fluorescent dye H_2_DCF-DA (Figure 2A). The measured fluorescent intensity was significantly higher in the G361-shCYGB cell line, which implies that CYGB knockdown leads to augmented levels of intracellular ROS. Following these findings, the sensitivity to RSL3 was assessed. RSL3 treatment resulted in a decrease in cell viability in both cell lines (Figure 2B). However, cell viability in G361-shCYGB was significantly lower compared to the control. Furthermore, the level of lipid peroxidation occurring 4 h after RSL3 treatment was measured. Although RSL3 induced lipid peroxidation in both G361-shCTR and G361-shCYGB, lipid peroxidation was noticeably higher in CYGB knockdown cells (Figure 2F; see also Appendix A).

As ferroptosis is regulated by the transcription factor nuclear factor erythroid 2-related factor 2 (NRF2) and its downstream targets, expression levels of NRF2 and HO-1 were measured. RSL3 treatment greatly induced HO-1 mRNA expression but only slightly increased NRF2 levels (Figure 2C). No significant differences in mRNA levels between G361-shCTR and G361-shCYGB were observed. Immunoblotting experiments showed, however, that both NRF2 and HO-1 protein levels were dramatically increased in both cell lines after RSL3 treatment (Figure 2D,E). Additionally, CYGB protein expression was slightly elevated in both G361-shCTR and G361-shCYGB. 

Of special note, a clear difference in the basal expression of NRF2 and HO-1 was observed, with protein levels being significantly decreased in CYGB knockdown G361-shCYGB cells (Figure 2E; see also Appendix A). However, RSL3 treatment led to a significantly higher fold change in HO-1 protein levels in G361-shCYGB cells compared to G361-shCTR. No difference was observed in NRF2 protein induction between G361-shCTR and G361-shCYGB cells.

### 3.3. CYGB-Dependent Antioxidant Response Element-Driven Luciferase Activity Is Increased upon RSL3 Treatment and NRF2 Overexpression

To obtain additional independent support of the NRF2-dependent regulation of the antioxidant response through binding to the antioxidant response element (ARE), a reporter assay using an ARE-driven luciferase gene was employed. Induction of ARE-driven luciferase activity was assessed under RSL3-treated conditions (Figure 3). RSL3 treatment clearly increased luciferase activity in both G361 control and CYGB knockdown cells compared to the untreated control. A small tendency towards increased activity was also seen under RSL3-treated conditions in CYGB-deficient cells versus control cells. Overexpression of NRF2 profoundly increased luciferase activity even further under both in RSL3-treated and untreated cells. Moreover, the measured luciferase activity was mostly increased RSL3-treated NRF2-overexpressing G361-shCTR and G361-shCYGB cells. Interestingly, ARE-driven luciferase activity was significantly higher in CYGB knockdown cells compared to G361-shCTR NRF2 overexpression cells, and substantially elevated in RSL3-treated cells (Figure 3). However, under basal conditions, luciferase activity was significantly lower in G361-shCYGB compared to G361-shCTR cells.

### 3.4. RNA Sequencing Analysis of RSL3-Treated G361-shCTR and G361-shCYGB versus Control

As the treatment of G361-shCTR and G361-shCYGB cells with RSL3 resulted in a different degree of lipid peroxidation and cell viability (Figure 2), we explored the CYGB-dependent transcriptome in G361-shCTR and G361-shCYGB cells under basal versus RSL3-treated conditions. Differential analysis of normalized counts using DESeq2 identified 316 genes that were differentially expressed under basal conditions. Of those 316 genes, 111 genes were differentially expressed above an absolute log_2_ fold change of 1 (DEGs) (see [31] for more details).

Moreover, to identify enriched sets of genes between the G361-shCTR and G361-shCYGB datasets, fGSEA was next performed (Figure 4A). fGSEA analysis showed that multiple hallmarks were positively enriched in G361-shCYGB, including cell cycle-related hallmarks mitotic spindle, G2M checkpoint and E2F targets, as well as cancer-associated hallmarks hedgehog signaling, IL2-STAT5 signaling, inflammatory response, estrogen response early, PI3K-AKT-mTOR, KRAS signaling, TNFα signaling via NF-κB and epithelial–mesenchymal transition. The hallmark-representing genes that are downregulated in response to ultraviolet (UV) radiation were also positively enriched. Additionally, several pathways were found to be negatively enriched in G361-shCYGB. Of those hallmark pathways, oxidative phosphorylation, fatty acid metabolism and cholesterol homeostasis are related to metabolism. The remaining pathways MYC targets, angiogenesis, Wnt/β-catenin signaling and KRAS signaling are associated with cancer.

To further gain insight into biological processes, the 111 significantly DEGs were clustered by their gene ontology (GO) and the 40 most differentially expressed ontology terms were plotted (Figure 4B). Some of the top GO terms enriched are involved in response to drugs, cell division, proliferation, migration and differentiation, but also calcium and sodium ion transport and the immune response to other organisms.

Upon RSL3 treatment, 8939 genes were found to be differentially expressed in G361-shCTR. Compared to the untreated control, a total of 2980 significantly DEGs, of which 2034 were upregulated and 946 downregulated, were identified (Figure 5A). fGSEA analysis showed that 25 hallmark pathways were significantly enriched in the RSL3-treated group (Figure 6A). RSL3 treatment seemed to induce multiple signaling pathways, including TNFα signaling via NF-κB, IL6-JAK-STAT, IL2-STAT, MTORC1 and KRAS signaling. Furthermore, treatment enriched several stress-related pathways, i.e., hypoxia, apoptosis, the P53 pathway, the UV response (up and down), the reactive oxygen species pathway and the unfolded protein response. Hallmark pathways E2F targets, oxidative phosphorylation, G2M checkpoint, MYC targets, fatty acid metabolism and DNA repair were negatively correlated with RSL3 treatment. GO enrichment analysis of the DEGs assigned these genes to be involved in processes related to the regulation of transcription, apoptosis, proliferation, and the inflammatory response (Figure 6C). Furthermore, GO terms encompassing signaling, e.g., MAPK cascade, phosphatidyl-inositol-3 kinase (PI3K) and circadian rhythm, were also found to be over-represented among others.

DESeq2 analysis of RSL3-treated G361-shCYGB identified 9574 differentially expressed genes (Figure 5B). In total, 3597 matched the criteria of DEGs, of which 2451 and 1146 were up- and downregulated, respectively. fGSEA analysis of the G361-shCYGB datasets showed a very similar pattern of significantly enriched hallmark pathways to that of G361-shCTR treated cells (Figure 6B). Of the pathways that were positively enriched in the RSL3-treated group, only the hallmark reactive oxygen species pathway was not significantly enriched compared to the fGSEA analysis of G361-shCTR. Hallmark pathways G2M checkpoint, mitotic spindle-related, oxidative phosphorylation and E2F targets were negatively enriched in RSL3-treated G361-shCYGB cells. In concordance with the fGSEA analysis, GO enrichment analysis of the DEGs showed very similar enrichment of GO terms compared to G361-shCTR cells (Figure 6D).

A ferroptosis gene signature was clearly present in both G361-shCTR and G361-shCYGB upon RSL3 treatment (Appendix A).

### 3.5. Differences between RSL3-Treated G361-shCTR and G361-shCYGB Cells

Finally, we investigated whether the knockdown of CYGB resulted in differences in the response to RSL3 treatment. Therefore, we compared the RSL3-treated G361-shCTR and G361-shCYGB datasets. DESeq2 analysis identified 1461 genes with altered gene expression, of which 354 (240 up- and 114 downregulated) were significant DEGs (Figure 7A). fGSEA analysis showed that hallmark pathways TNFα signaling via NF-κB, inflammatory response, hypoxia, P53 pathway, UV response (up), IL2-STAT5 signaling, coagulation, KRAS signaling (up), allograft rejection, IFNγ response, epithelial–mesenchymal transition, myogenesis, estrogen response, MTORC1 signaling, and oxidative phosphorylation were all enriched in RSL3-treated G361-shCYGB cells compared to G361-shCTR. In contrast, hallmarks G2M checkpoint, mitotic spindle and Wnt/β-catenin signaling were rather enriched in RSL3-treated G361-shCTR cells (Appendix A). GO enrichment analysis of the 354 DEGs showed (among others) an overrepresentation of GO terms related to signal transduction, transcription, the inflammatory response, the extracellular matrix and IL1β production (Figure 7B).

Interestingly, multiple pyroptosis-associated DEGs were found to be upregulated in G361-shCYGB cells compared to G361-shCTR upon RSL3 treatment (Figure 7B; see also Appendix A). We therefore validated the differential expression of five pyroptosis-related genes (CASP1, CD74, CXCR4, NLRP3, SYK) by RT-qPCR (Figure 7C).

## 4. Discussion

CYGB displays substantial sequence similarities with other globins and even has ~40% homology with myoglobin (Mb) [32]. It is now well established that mitochondria and Mb are intimately linked through their functional regulation of one another [33,34]. Considering the homology to Mb and their very similar oxygen affinity, we sought to investigate a potential effect of CYGB knockdown in G361 melanoma cells on mitochondrial respiration. Mitochondrial oxidative phosphorylation was significantly diminished in CYGB-depleted G361-shCYGB cells (Figure 1B–F), consistent with the RNA-seq data, where G361-shCTR cells were compared to G361-shCYGB cells under basal conditions. Moreover, fGSEA analysis identified three metabolism-related pathways that were significantly enriched in the G361-shCTR cells, of which the hallmark pathway oxidative phosphorylation was most enriched (Figure 4A).

CYGBs’ tumor-suppressive role has previously been investigated [17,35]. Several studies have shown that CYGB can function as a cytoprotective protein through the scavenging of ROS [12,14,36]. Recently, Zweier et al. demonstrated the potent superoxide dismutase activity of CYGB. In the current study, increased levels of ROS were found in G361-shCYGB cells under basal conditions, which would support an ROS-scavenging function (Figure 2A). Additionally, G361-shCYGB cells treated with the ferroptosis inducer RSL3 were significantly more sensitive and displayed increased levels of lipid peroxidation compared to treated G361-shCTR cells (Figure 2B,F). The increase in basal ROS levels in G361-shCYGB would facilitate the initiation of phospholipid peroxidation and subsequently lead to more ferroptotic cell death. Furthermore, CYGB has been shown to reduce anionic phospholipids through innate peroxidase activity [13]. Therefore, the knockdown of CYGB would result in a diminished ability to clear phospholipid hydroperoxides (PLOOHs), and thus increase lipid peroxidation. However, the lack of increased lipid peroxidation observed under basal conditions in G361-shCYGB cells would argue against peroxidase activity.

The small molecule RSL3 induces ferroptosis by directly inhibiting glutathione peroxidase 4 (GPX4) [36]. GPX4, a selenoprotein, is the major enzyme catalyzing the reduction, and thus toxification, of PLOOHs in mammalian cells, which makes it an essential regulator of ferroptotic cell death. GPX4 is a downstream target of the master regulator of the antioxidant response NRF2. Furthermore, NRF2 target genes have been shown to regulate the activity of proteins and enzymes responsible for preventing lipid peroxidation, including the enzyme HO-1. Whereas only NRF2 protein expression was stabilized upon RSL3 treatment in both cell lines, HO-1 was clearly induced both on the mRNA and protein level (Figure 2C–E). NRF2’s activity and abundance are tightly regulated at the transcriptional, post-transcriptional and post-translational level [37,38]. On the other hand, NRF2 itself, as a redox-sensitive transcription factor, regulates the expression of antioxidant and detoxifying genes, including HO-1, through binding to the cis-acting enhancer sequence ARE. Consistently, ARE-driven reporter assays demonstrated increased luciferase activity upon RSL3 treatment under basal and NRF2 overexpression conditions in CYGB-deficient G361 cells (Figure 4).

Intriguingly, the knockdown of CYGB also affected NRF2 and HO-1 protein expression under basal conditions (Appendix A). This is in accordance with previous studies [14,39,40,41] and is consistent with results obtained recently by our group [31]. Considering NRF2’s central role in heme biosynthesis, and taking into account CYGB as a heme-containing protein, cells possessing abundant endogenous CYGB levels constitutively induce NRF2 protein, and subsequently HO-1 [42,43]. Furthermore, CYGB, as a hemeprotein, could potentially be involved in the regulation of the labile iron pool, and thus ameliorate RSL3-mediated ferroptosis.

Notably, HO-1 protein levels were significantly higher in G361-shCYGB cells upon RSL3 treatment. There is conflicting evidence concerning the role of HO-1 in ferroptosis, suggesting HO-1 to either promote or suppress ferroptosis [44,45]. However, it seems that the pro- or anti-ferroptotic effect of HO-1 is dependent on the intracellular ROS levels and degree of HO-1 induction [46]. HO-1 breaks down heme into biliverdin/bilirubin, carbon monoxide and ferrous iron. Although biliverdin and bilirubin are antioxidants, ferrous iron is highly reactive and can lead to the accumulation of ROS. Therefore, labile free iron is sequestered by ferritin. Under moderate levels of HO-1 induction, the labile free iron is adequately buffered by ferritin and therefore HO-1 can exert its cytoprotective effect. However, higher levels of HO-1 expression could exceed the labile free iron buffering capacity of the cell, thereby facilitating lipid peroxidation and subsequent ferroptotic cell death [43]. Thus, the increased induction of HO-1 in CYGB knockdown G361-shCYGB cells, together with the increased basal ROS levels, could potentially have contributed to the increased sensitivity towards RSL3 treatment.

Another regulator of ferroptosis that recently was discovered, ferroptosis-suppressor protein 1 (FSP1; also known as AIFM2), plays a prominent role in the suppression of ferroptosis [47,48]. FSP1 suppresses lipid peroxidation through the reduction of ubiquinone to ubiquinol, which in turn either directly reduces lipid radicals to terminate lipid autoxidation, or indirectly via regenerating vitamin E. In T cell lymphoblastic lymphoma cells, FSP1 was shown to be upregulated by the long non-coding RNA maternally expressed 3 (MEG3), a regulatory RNA involved in tumor development [49].

Interestingly, MEG3 was one of the DEGs identified to be dramatically downregulated in G361-shCYGB cells upon RSL3 treatment (Figure 7A) and was recently found to also be downregulated under basal conditions [31]. Furthermore, MEG3 was shown to inhibit tumor formation, growth and metastasis in melanoma [50]. Thus, whilst the knockdown of CYGB in G361 melanoma cells gives rise to a more malignant cancer phenotype, it also makes cells more susceptible to ferroptosis. Additionally, multiple cancer-associated pathways were enriched in the CYGB-deficient G361-shCYGB cell line compared to G361-shCTR. Overactivation of the PI3K/AKT/mTOR pathway is known to promote proliferation and survival [51]. Oncogenic KRAS promotes cell survival, proliferation and cytokine secretion [52]. One of the cytokines secreted is TNFα, which in turn activates NF-κB, which promotes tumor metastasis and invasiveness [53]. Hedgehog signaling has been shown to promote metastasis through its involvement in the epithelial–mesenchymal transition (EMT) [54]. EMT transforms polarized epithelial cells into motile mesenchymal cells, thereby facilitating invasiveness and metastasis [55]. Thus, the knockdown of CYGB seems to induce cellular changes that lead to a more malignant cancer phenotype, further supporting a tumor-suppressive role for CYGB. 

Remarkably, multiple DEGs found to be upregulated in G361-shCYGB cells compared to G361-shCTR upon RSL3 ferroptosis treatment are associated with pyroptosis (Figure 7B). More specifically, NOD-, LRR- and pyrin domain-containing protein 3 (NLRP3) and caspase 1 (CASP1) were significantly upregulated, as confirmed by RT-PCR (Figure 7C). Pyroptosis is driven by inflammatory caspases that are activated upon the formation of a multiprotein complex called the inflammasome [56,57]. The NLRP3 inflammasome consists of a sensor (NLRP3), an adaptor (ASC; also known as PYCARD) and an effector (caspase 1). NLRP3 and CASP1 transcription is upregulated by the activation of pathogen-associated molecular patterns (PAMPS), toll-like receptors (TLRs) or cytokines such as TNFα and IL-1β that lead to NF-κB activation and gene transcription [58,59], which is perfectly in line with the findings here, showing the enrichment of hallmark pathway TNFα via NF-κB signaling and GO term IL-1β production (Figure 7B and Appendix A). Additionally, cluster of differentiation (CD74) together with C-X-C chemokine receptor type 4 (CXCR4) were also upregulated. CD74 is involved in cell signaling by acting as a receptor for the pro-inflammatory cytokine macrophage migration inhibitory factor (MIF) [60]. However, to trigger the intracellular signal transduction, the presence of co-receptor protein CXCR4 is needed [61]. MIF can initiate several cellular signaling pathways in a CD74-dependent manner, including signaling cascades involving Syk and NF-κB [62,63,64], and promotes the expression of TLR4 [65]. Moreover, accumulating evidence suggests that MIF plays a central role in NLRP3 inflammasome activation [66,67].

Collectively, our results revealed a cytoprotective role for CYGB in suppressing ferroptosis and pyroptosis cell death hallmarks. Moreover, transcriptome analysis revealed the enrichment of multiple cancer-associated pathways in CYGB knockdown G361 cells, demonstrating the tumor-suppressive function of CYGB. This could potentially have important therapeutic implications. Meta-analysis of the whole-genome bead array dataset (GSE29359) comparing human primary melanocyte cell lines and 82 patient-derived metastatic melanoma samples demonstrated that although CYGB is mostly lost during melanocyte-to-melanoma transition, some melanoma cells retain their high endogenous CYGB levels [16]. Therefore, modulated CYGB expression during melanomagenesis can determine the sensitivity towards treatments focused on inducing ferroptosis. CYGB expression in (metastatic) melanoma tissues could be used as a biomarker for determining the best therapeutic approach and treatment outcome.

Remarkably, RSL3 treatment in G361-shCYGB led to the activation of the NLRP3 inflammasome and subsequent induction of pyroptosis. Several studies already proposed an anti-inflammatory role for CYGB through the regulation of the cytokine TNFα [68,69] via inhibition of the NF-κB pathway [70], which is in accordance with our RNA-seq data (Figure 2). CYGB was also shown to inhibit LPS-induced NADPH oxidase activity and ROS, NO and O_2_^•−^ generation [71]. CYGB could possibly regulate inflammasome activation by attenuating inflammation through its polyvalent RONS-scavenging functions. However, cardiolipin (CL) was shown to be necessary for NLRP3 inflammasome activation [72]. As CYGB can interact with and regulate the redox status of CL [41,73], CYGB could indirectly regulate pyroptotic cell death.

Conflicting evidence exists regarding the tumor-promoting or -inhibiting role of pyroptosis [74]. However, inducing pyroptosis in tumor cells holds great promise as a novel cancer treatment strategy. Here, pyroptosis likely contributed to the increased overall cell death in CYGB knockdown G361 cells. Studies focused on pyroptosis signaling pathways would provide opportunities for finding tumor biomarkers and novel chemotherapeutic drug development [75]. In this light, CYGBs could be of great interest for further research.

Of special note, changes in mitochondrial aldehyde dehydrogenase metabolism have been demonstrated at the crossroad of ferroptosis and pyroptosis signaling pathways [76]. Along the same line, Kang et al. recently reported that the key ferroptosis regulator GPX4 also acts as a negative regulator of the pyroptotic cell death pathway [77]. Furthermore, pyroptosis was also shown to be induced by increased intracellular iron and ROS in melanoma [78]. Hence, RSL3, as a GPX4 inhibitor, can lead to pyroptosis.

## 5. Conclusions

In this study, we explored the role of CYGB in determining the cellular sensitivity towards RSL3-mediated ferroptosis. The presence of the cytoprotective protein CYGB determined the outcome of RSL3 treatment through the regulation of lipid peroxidation and ROS levels, supporting a redox-regulatory role. RSL3 treatment strongly increased the expression of the master regulator of the antioxidant response NRF2 and downstream target HO-1, whereas CYGB protein levels determined the basal expression of NRF2 and HO-1, likely mediated through MEG3. Moreover, transcriptome analysis following CYGB knockdown further revealed the enrichment of multiple cancer malignancy pathways, supporting a tumor-suppressive function of CYGB. Remarkably, RSL3 treatment of G361-shCYGB cells also led to the activation of the NLRP3 inflammasome and pyroptosis pathways, which identified CYGB expression regulation as a critical determinant of the ferroptosis–pyroptosis therapy response. CYGB could be an exciting new predictive biomarker candidate.

## Figures and Tables

**Figure 1 antioxidants-11-01548-f001:**
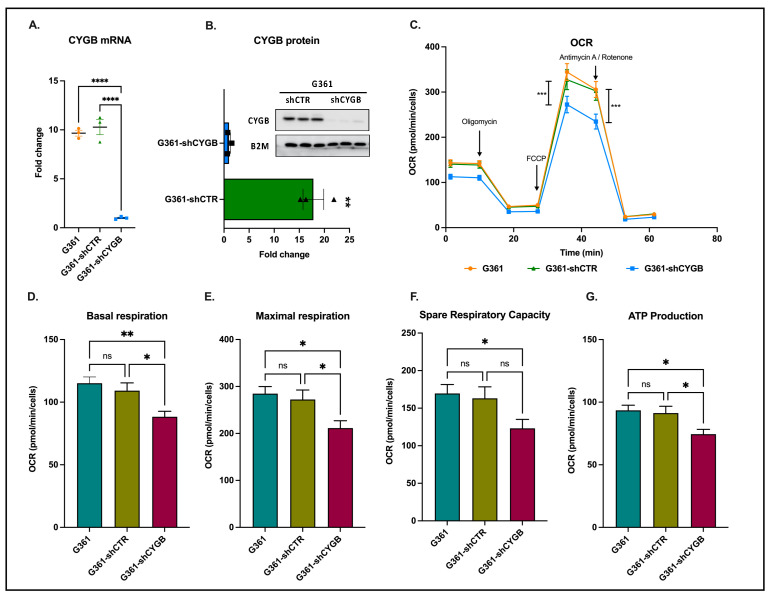
CYGB knockdown reduces mitochondrial respiration. Knockdown of CYGB in G361 cells was validated at mRNA (**A**) and protein (**B**) level. (**C**) The average oxygen consumption rate (OCR) was measured between G361 (untransduced), G361-shCTR (transduction control) and G361-shCYGB (CYGB knockdown). (**D**) Initially, the basal respiration rate was measured. (**E**) After oligomycin injection, ATP production was measured. (**F**) Subsequently, the maximal respiration was measured, after FCCP injection. (**G**) The spare respiratory capacity was measured upon addition of antimycin A and rotenone. Results are depicted as the mean with S.E.M. of three independent experiments (n = 3). One-way ANOVA (**A**), Student *t*-test (**B**), two-way ANOVA (**C**–**G**) (* *p* ≤ 0.05; ** *p* ≤ 0.01; *** *p* ≤ 0.001; **** *p* ≤ 0.0001, ns: non-significant). FCCP: Carbonyl cyanide-4 (trifluoromethoxy) phenylhydrazone.

**Figure 2 antioxidants-11-01548-f002:**
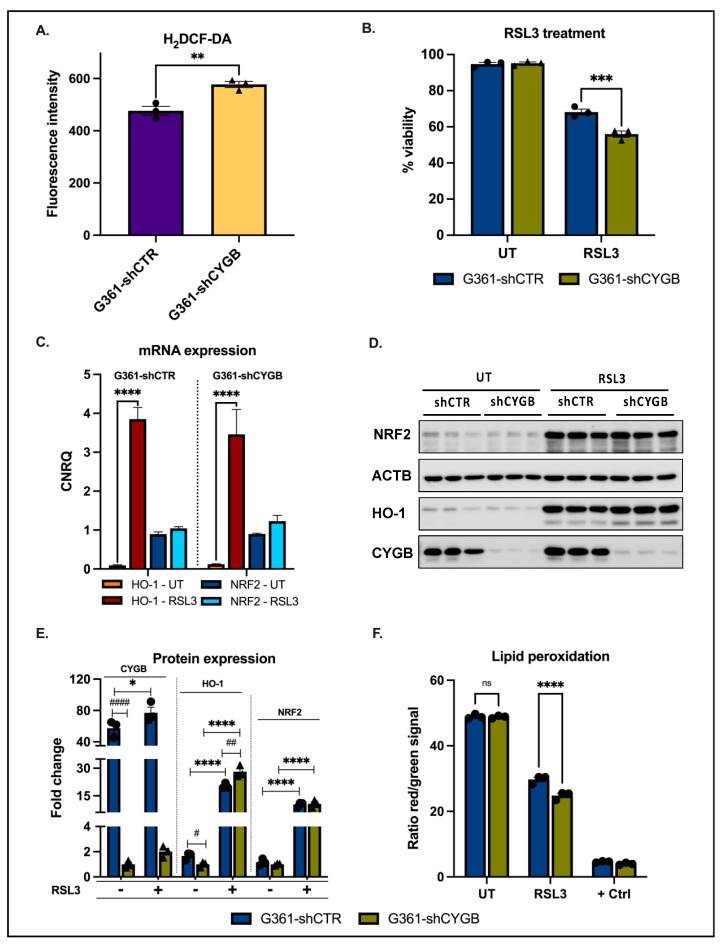
Cytoprotective effect of CYGB. (**A**) Knockdown of CYGB increased basal intracellular ROS levels. (**B**) G361-shCTR and G361-shCYGB cells were treated with 7.5 μM RSL3 and cell viability was measured after 24 h using propidium iodide (PI) staining. (**C**) The average calibrated normalized relative quantities (CNRQ) of HO-1 and NRF2 mRNA under basal conditions (UT) or upon RSL3 treatment. CNRQ values were normalized to B2M and YWHAZ. (**D**) Immunoblotting results of NRF2, HO-1 and CYGB 6h after RSL3 treatment. ACTB was used as loading control. (**E**) The average fold change in protein expression of HO-1, NRF2 and CYGB in G361-shCTR and G361-shCYGB cells, compared to the untreated G361-shCYGB samples (set as 1). Measured immunoblot signals were normalized to the loading control ACTB. (**F**) Average ratio of the measured red (reduced) over green (oxidized) signal of the BODIPY 581/591 C11 reagent in G361-shCTR and G361-shCYGB cells. Cells were either untreated or treated with 7.5 μM RSL3 or 100 μM cumene hydroperoxide (positive control). All results are depicted as the mean with S.E.M. of three independent experiments (n = 3). (**A**) Student’s *t*-test (** *p* ≤ 0.01), (**B**–**F**) two-way ANOVA (*^/#^ *p* ≤ 0.05; **^/##^ *p* ≤ 0.01; *** *p* ≤ 0.001; ****^/####^ *p* ≤ 0.0001, ns: non-significant).

**Figure 3 antioxidants-11-01548-f003:**
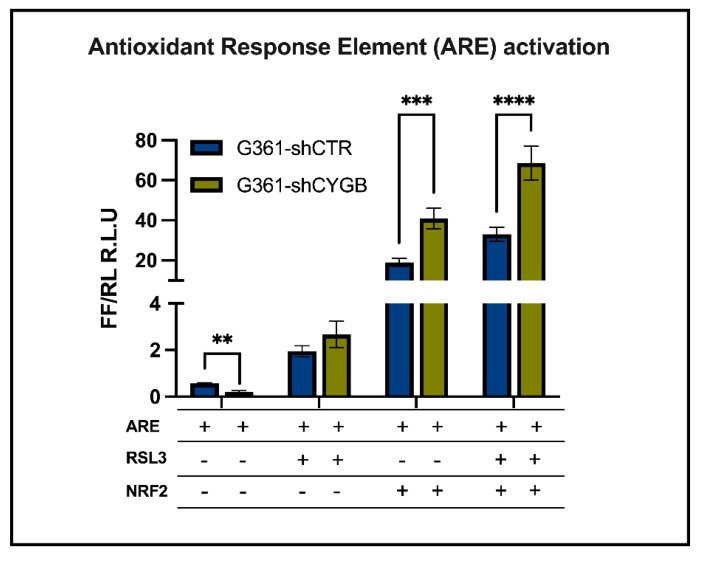
CYGB-dependent changes in antioxidant response element-driven luciferase activity. Binding of NRF2 to the antioxidant response element (ARE) in G361-shCTR and G361-shCYGB cells was assessed using an ARE-driven luciferase gene reporter assay. Cells were co-transfected with *Renilla* luciferase reporter vector to control for differences in transfection efficiency and extract preparation. Reporter activities were expressed as relative firefly/*Renilla* luciferase activities (R.L.U.). Luciferase reporter activity was measured after 6 h under basal conditions or upon 10 μM RSL3 treatment in the presence or absence of NRF2 overexpression in G361-shCTR and G361-shCYGB cells. Results are depicted as the mean with S.E.M. of six independent experiments (n = 6). Two-way ANOVA (** *p* ≤ 0.01; *** *p* ≤ 0.001; **** *p* ≤ 0.0001).

**Figure 4 antioxidants-11-01548-f004:**
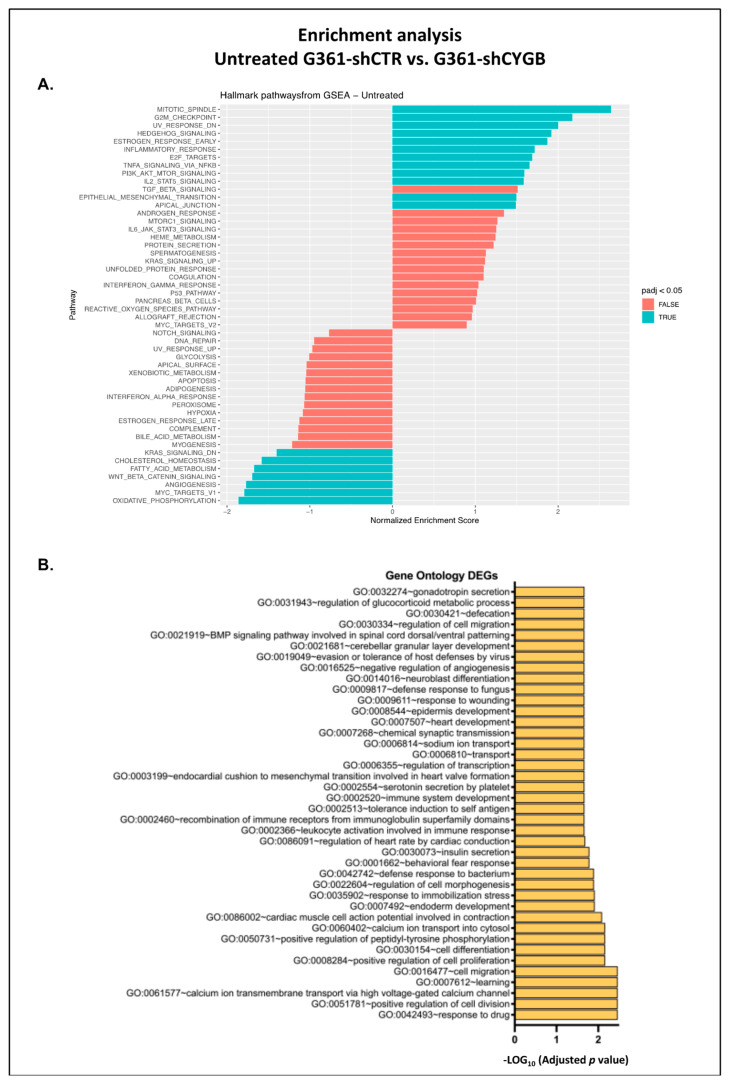
Basal CYGB-dependent transcriptome. Comparison of the G361-shCTR and G361-shCYGB transcriptomes under basal conditions using (**A**) fast Gene Set Enrichment Analysis (fGSEA), using the hallmark pathway gene sets and (**B**) gene ontology (GO) enrichment analysis.

**Figure 5 antioxidants-11-01548-f005:**
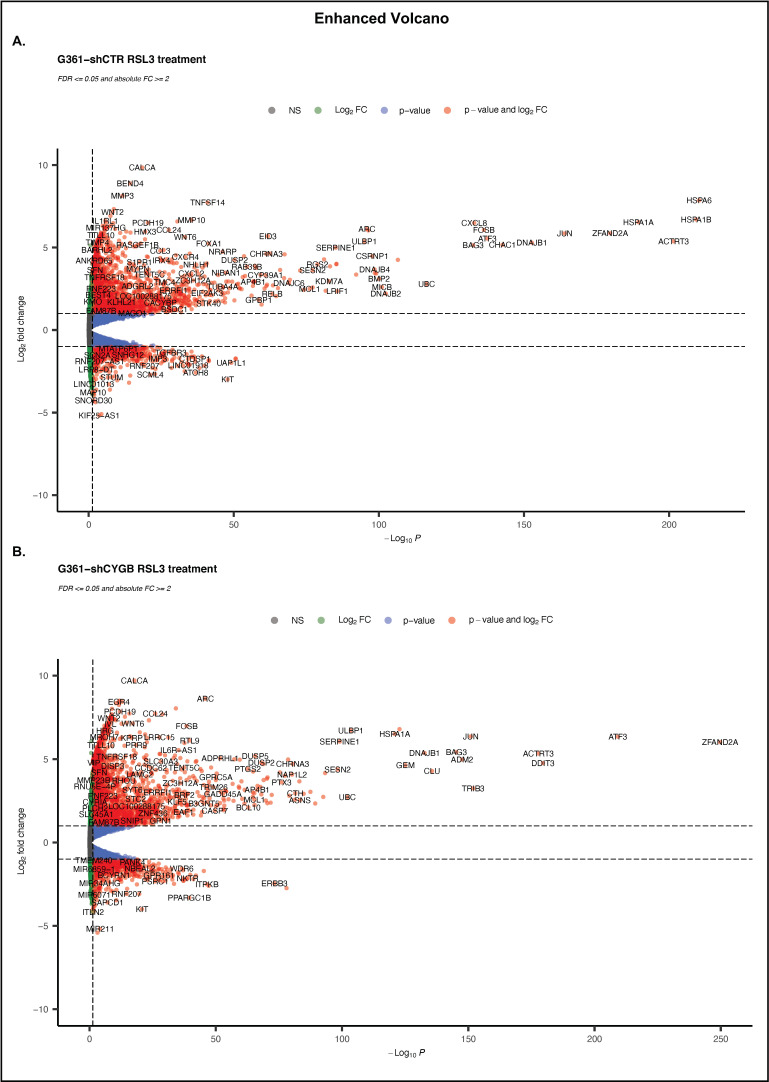
Enhanced volcano RSL3 treatment. Volcano representation of the differential expression analysis (DESeq2) of RSL3-treated G361-shCTR (**A**) and G361-shCYGB cells (**B**). Differentially expressed genes were represented based on their Log_2_ fold change (Log_2_FC) and −Log_10_ *p*-adjusted value (-Log_10_
*P*).

**Figure 6 antioxidants-11-01548-f006:**
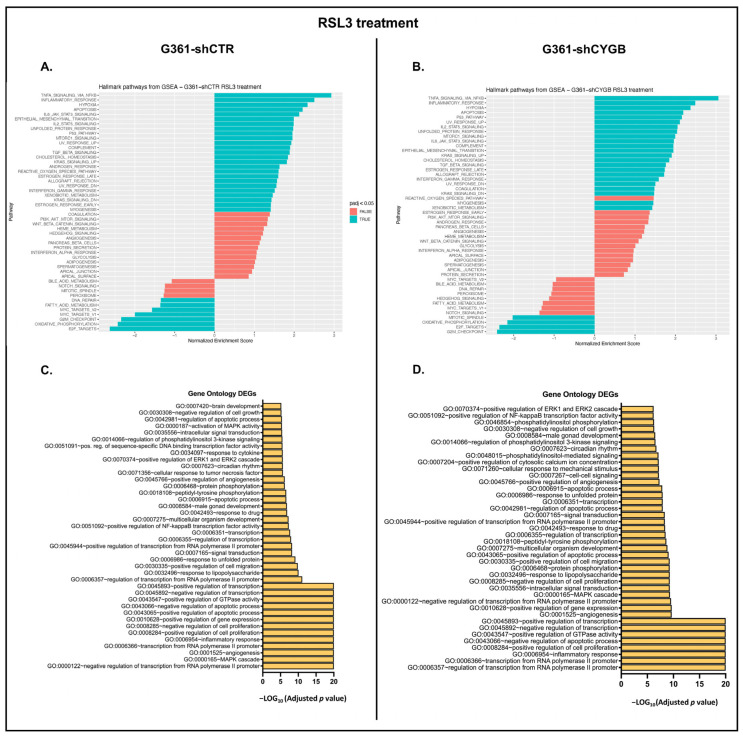
RSL3-mediated transcriptomic changes. Analysis of the transcriptomic changes induced by RSL3 treatment in G361-shCTR and G361-shCYGB cells compared to the untreated control. (**A**,**B**) fast Gene Set Enrichment Analysis (fGSEA) using the hallmark pathway gene sets was performed on G361-shCTR (**A**) and G361-shCYGB (**B**) melanoma cells. Gene ontology (GO) enrichment analysis was performed on the significantly differentially expressed genes (DEGs) in G361-shCTR (**C**) and G361-shCYGB (**D**) cells.

**Figure 7 antioxidants-11-01548-f007:**
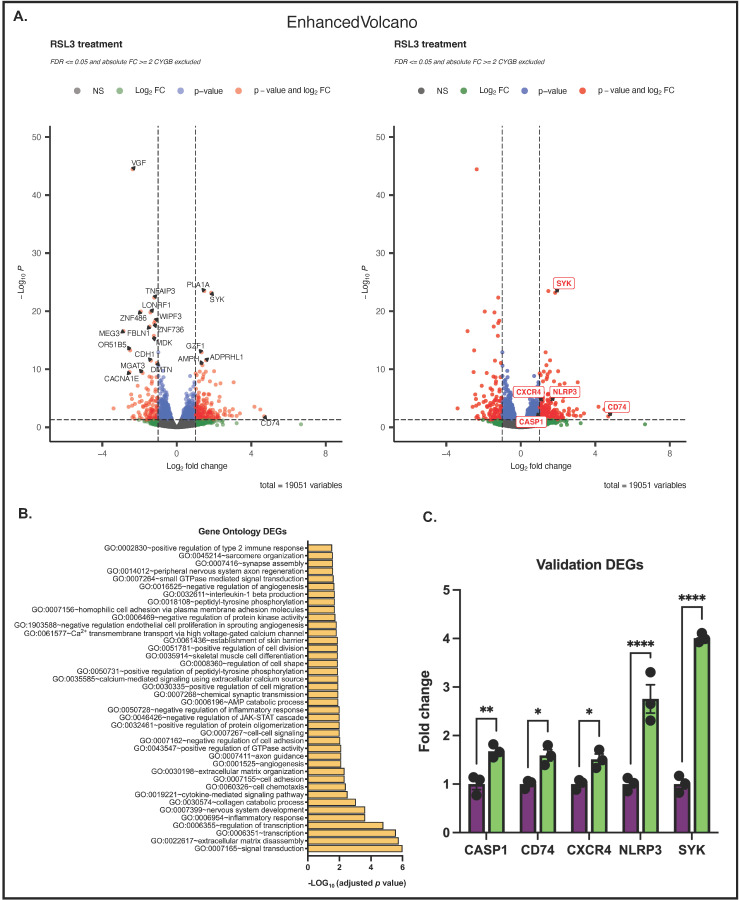
CYGB-dependent differences in response to RSL3 treatment. Comparison of RSL3-treated G361-shCTR and G361-shCYGB cells. (**A**) Enhanced volcano plots of the differential expression analysis (DESeq2) of RSL3-treated G361-shCTR versus G361-shCYGB cells. Differentially expressed genes were represented based on their Log_2_ fold change (Log_2_FC) and −Log_10_ *p*-adjusted value (−Log_10_
*P*). (**B**) The top-ranked gene ontology (GO) terms of the differentially expressed genes (DEGs) identified upon GO enrichment analysis. (**C**) Validation of selected DEGs by RT-PCR. Fold change represents the average fold change expression in G361-shCYGB cells compared to G361-shCTR cells (set as 1). Results are depicted as the mean with S.E.M. of three independent experiments (*n* = 3). One-way ANOVA (* *p* ≤ 0.05; ** *p* ≤ 0.01; **** *p* ≤ 0.0001).

## Data Availability

The data are contained with this article and Appendix A.

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
