# Peer review of "Cytoglobin Silencing Promotes Melanoma Malignancy but Sensitizes for Ferroptosis and Pyroptosis Therapy Response"

_antioxidants, 2022, doi:10.3390/antiox11081548_

Round 1

Reviewer 1 Report

In my opinion, in the presented form the manuscript (antioxidants-1842961) entitled ‘Cytoglobin silencing promotes melanoma malignancy but sensitizes for ferroptosis and pyroptosis therapy response’ described by Joey De Backer, Darko Maric, Karim Zuhra, Annemie Bogaerts, Csaba Szabó, Wim Vanden Berghe, David Hoogewijs can be recommended for publication in Antioxidants after minor revision.

My remarks and recommendations to the Antioxidants are as follows:

The text is comprehensible.

The studies presented by the authors are interestingly described, properly and reliably documented by the novelty results. These results of the experiments regarding to in melanoma treatment are presented, described and commented on at a high scientific level.

After reading the publication, I have doubts about the scientific novelty of the scientific research of the reviewed manuscript. Therefore, the Authors should clearly indicate what is the scientific novelty of their research. Please indicate clearly what is new with your manuscript (e.g., Conclusions) for the Antioxidants, especially in comparison to earlier of publication(s).

Author Response

Dear reviewer, please see the attachment for a point-by-point response.

Reviewer 2 Report

Authors in their study have shown the importance CYGB as a modulator of ferroptosis and pyroptosis cell death signalling pathways in melanoma cells. This study is of high interest and very well written. However, I have few comments which should be addressed to make the study more interesting, as listed below:

1.) Authors in the introduction section stated "G361 melanoma cells which express highly abundant endogenous levels of CYGB". No reference or justification is provided for such a claim. Authors need to provide the citation for Yoshihiko Fujita et al 2014. In this article authors used a panel of 18 cell lines and found G361 to possess maximum mRNA levels of CYGB

2.) Authors stated cells were treated with 7.5 μM RSL3. But I could not understand the reason for such a specific concentration. Was this concentration & treatment point adopted from previous studies? If yes, then cite. If no, please provide the dose kinetics as a supplementary figure.

3.) Please provide the representative image of Fig 2F (BODIPY) in supplementary section for better visualization & perception for reader.

4.) In result section, line 356-357, authors stated "Of those 316 genes, 111 genes were differentially expressed above an absolute log2 fold change of 1 (DEGs) (De Backer et al. in press)." It is a bit confusing, Why Da backer et al is referenced here when it is in press & how does it correlates with the present manuscript. Please clarify.

5.) The discussion section is weak in terms of translational perspective of the study. I feel that the scientific point raised in introduction was totally missing in discussion section. It revolves only around the RNA seq data. Authors need to discuss more about the outcome of the study in terms of clinical settings in future. In my perspective, Anti-inflammatory role for CYGB should not be exaggerated too much as it could be due to treatment effect.

6.) Study was performed on a single cell line (although very appropriate to use G361), impact is somehow diluted. Authors need to perform Figure 1 E-G (OCR data) in either C32TG or P22 cell line, both of which have significantly higher CYGB expression.

Minor Comments:

1.) City & country code are not provided for the material used to perform experiments in Materials & Methods section.

2.) Amount of RNA (microgram or nanogram) used for reverse transcription before real-time PCR missing

3.) There are minor grammatical typo errors. Please proof read & correct them.

4.) In line 602, "visualization J.D.BX.X." what is XX. Please remove it.

Author Response

(The authors gave the same response as above.)

Round 2

Reviewer 2 Report

I would like to appreciate & congratulate authors for successfully answering and correcting all my concerns.

I have no more concern or question. Manuscript could be accepted for publication.